# Enzymatic Electroanalytical Biosensor Based on *Maramiellus colocasiae* Fungus for Detection of Phytomarkers in Infusions and Green Tea Kombucha

**DOI:** 10.3390/bios11030091

**Published:** 2021-03-22

**Authors:** Érica A. Batista, Giovanna N. M. Silva, Livia F. Sgobbi, Fabio B. Machado, Isaac Y. Macedo, Emily K. Moreno, Jerônimo R. Neto, Paulo S. Scalize, Eric S. Gil

**Affiliations:** 1Faculdade de Farmácia (FF), Universidade Federal de Goiás (UFG), Goiânia 74605-170, Brazil; eriquitaso2@gmail.com (É.A.B.); giovannamellonutri@gmail.com (G.N.M.S.); famafarm@yahoo.com.br (F.B.M.); isaacyvesl@gmail.com (I.Y.M.); milykussmaul@gmail.com (E.K.M.); jeronimoneto8@gmail.com (J.R.N.); 2Instituto de Química (IQ), Universidade Federal de Goiás (UFG), Goiânia 74690-900, Brazil; livia.fsgobbi@gmail.com; 3Escola de Engenharia Civil e Ambiental (EECA), Universidade Federal de Goiás (UFG), Goiânia 74605-220, Brazil; pscalize.ufg@gmail.com

**Keywords:** quality control, polyphenoloxidase enzymes, kombucha, green tea, electrochemical biosensor

## Abstract

In this work, we developed an enzymatic voltammetric biosensor for the determination of catechin and gallic acid in green tea and kombucha samples. The differential pulse voltammetry (DPV) methodology was optimized regarding the amount of crude enzyme extract, incubation time in the presence of the substrates, optimal pH, reuse of the biosensor, and storage time. Samples of green tea and kombucha were purchased in local markets in the city of Goiânia-GO, Brazil. High performance liquid chromatography (HPLC) and Folin-Ciocalteu spectrophotometric techniques were performed for the comparison of the analytical methods employed. In addition, two calibration curves were made, one for catechin with a linear range from 1 to 60 µM (I = −0.152 * (catechin) − 1.846), with a detection limit of 0.12 µM and a quantification limit of 0.38 µM and one for gallic acid with a linear range from 3 to 60 µM (I = −0.0415 * (gallic acid) − 0.0572), with a detection limit of 0.14 µM and a quantification limit of 0.42 µM. The proposed biosensor was efficient in the determination of phenolic compounds in green tea.

## 1. Introduction

Over the years, biosensor technology has drawn attention to the agri-food sector due to the high demand for analyses associated with contamination, adulteration, additives, allergens, and food packing problems in foodstuff [1,2]. In this scenario, electrochemical biosensors are attractive analytical tools that take advantage of remarkable attributes, such as easy-to-operate, sensitive, simple-to-construct, and fast responsive devices for monitoring safety and quality of food [3,4,5,6,7]. Among the components in foodstuffs, phenolic compounds have been highlighted since they provide added value, owing to their well-known health benefits and their function as stabilizers [8]. There is an increasing trend in the food industry regarding formulation enrichment with polyphenols for food and beverage. In the midst of polyphenol-rich drinks, teas deserve to be emphasized since they have been used for thousands of years, such as green tea (Camellia sinensis), a functional food known in traditional Chinese medicine. It has beneficial bioactive substances that assist in the reduction of body fat and systemic arterial hypertension, and it has antioxidant, hypoglycemic, anticholisteremic, hypolipidemic, and thermogenic activities [9].

Consumed mainly by infusion, decoction or fermentation, black tea and kombuchas have polyphenolic phytomarkers, predominantly flavonols and phenolic acids. The fermented form of green tea, kombucha, has high nutritional value through the fermentation mechanism of symbiotic culture of bacteria and yeasts (SCOBY), which generates antioxidant and beneficial health properties, such as water-soluble vitamins B1, B2, B6, and C, as well as amino acids and some minerals, optimizing the phenolic compounds of hydrolysis [10,11].

Currently, there are no quality indicators for green tea-based beverages present in the market, allowing adulteration by adding other condiments and teas with different properties such as mate tea, which can compromise the benefits and quality of nutrients. The fundamental difference between the production of green tea and mate tea is the drying method which leads to characteristic flavor and aroma compounds [12]. Therefore, polyphenolic compounds found in green tea contrast considerably with mate tea. For instance, green tea contains high concentrations of catechin and gallic acid whereas mate tea does not contain any of them [13]. Those compounds could act as fingerprint parameters for beverage quality.

Several electrochemical biosensors have been developed to monitor phenolic compounds using laccase enzyme (EC 1.10.3.2) which catalyzes mainly the oxidation of diphenols towards oxygen acting as an electron acceptor [14,15]. Laccase is an oxidoreductase found in plants and in fungi extracellular digestion in ligninocellulosic degradation and phenolic compounds such as catechol, catechins, and organic acids. Despite the potential of laccase-based biosensors as a sensitive analytical tool for phenolic compound detection, the commercial/industrial implementation of those devices is not available. This limitation is mostly due to laccase high production costs associated with isolation and purification processes [16].

As an alternative to commercial laccase, crude enzymatic extracts offer a viable possibility to be applied in electrochemical biosensors, since the natural enzymatic environment is preserved, which increases enzyme stability [17]. Moreover, cofactor and coenzymes may be present in the crude extract. White rot fungi are important phenolic degraders and have intense enzymatic activity, such as the pathogenic fungus *Marasmiellus colocasiae* [18,19,20]. Crude enzymatic extract from Marasmiellus colocasiae is rich in polyphenoloxidase allowing the development of electrochemical biosensors for phenolic compound detection [5].

In this work, we describe different methods to obtain total polyphenoloxidase (TPO) enzymes extracted from *Marasmiellus colocasiae* fungus for the development of carbon paste biosensors, in order to detect phenolic phytomarkers for quality control of natural and fermented products. The resulting optimized biosensor presented good reuse signal stability and electrocatalytic affinity.

## 2. Materials and Methods

### 2.1. Chemical Reagents

Methanol HPLC grade was supplied by J. T. Baker (Phillipsburg, NJ, USA). Phosphoric acid was obtained from Vetec (Rio de Janeiro, Brazil). All solutions were prepared using analytical grade reagents and double distilled Milli-Q water (conductivity ≤ 0.1 µS.cm^−1^) (Millipore S.A., Molsheim, France). Gallic acid, catechin, Folin-Ciocalteu reagent, 2,2′-azino-bis(3-ethylbenzothiazoline-6-sulfonic acid) diammonium salt (ABTS) were purchased from Sigma Chemical Co, (St. Louis, MO, USA). All solution standards were prepared with a concentration of 100 µM from the dilution of stock solutions (1 mM).

### 2.2. Origin and Maintenance M. colocasiae

The fungus *M. colocasiae* CCIBT 3388, isolated in Domingos Martins-ES (Brazil)/2005, was obtained from the Basidiomycete Cultures Collection (CCB) from the Institute of Botany (São Paulo, Brazil). The identification of the source of obtaining ex situ genetic heritage, with the information contained in the deposit record, was determined by § 1 of article 22 of the decree n° 8.772 of 2016.

### 2.3. Enzymatic Production and Growth

#### 2.3.1. Solid Culture Medium

The solid culture medium, known as MEA, was prepared mixing 1.5% agar (Biolog^®^, Hayward, CA, USA), 2% malt extract (Biolog^®^), 1% dextrose (Greentec^®^, Cambridge, ON, Canada), and 0.1% bacteriological peptone (Biolog^®^). For inoculation, a fungal disc 5 mm in diameter was placed in the center of the Petri dish containing MEA. Petri dishes were kept in an oven at 28 °C between 7 to 10 days, so that the mycelium occupied 3/4 of the medium surface.

The enzymatic extract was obtained from the MEA. The entire contents of a Petri dish were cut into squares and placed in a buffer solution of 50 mM sodium acetate pH 5.0. The solution with the solid mixture was homogenized manually for 3 min, followed by stirring at 120 rpm for 60 min, then, manually stirred for 3 min.

#### 2.3.2. Liquid Culture Medium

To obtain the liquid culture medium, called ME, 2% malt extract, 1% dextrose, and 0.1% bacteriological peptone were used. For inoculation, 15 fungal discs of approximately 5 mm in diameter, obtained from the MEA containing *M. colocosiae*, were added in 250-mL conical flasks containing 100 mL of liquid substrate. Subsequently, they were incubated and kept in an oven at 28 °C for 15 days.

#### 2.3.3. Liquid Culture Medium with Salt Addition

To obtain the liquid medium with added salts, called MES, the ME was used with addition of 0.250% copper sulfate, 0.01% magnesium sulfate, 0.06% manganese sulfate, 0.019% dipotassium phosphate, 1.8% soy, and 0.2% tween oil. For inoculation in the MES, 15 fungal discs of approximately 5 mm in diameter, obtained from the MEA containing *M. colocosiae*, were added in 250-mL conical flasks containing 100 mL of sterile liquid substrate. Subsequently, they were incubated and kept in an oven at 28 °C for 15 days.

All crude enzymatic extracts obtained by growth in the three different media were vacuum filtered in a filtration system with 0.45-µm cellulose nitrate membrane and 47 mm in diameter (Unifil^®^). The enzymatic extracts were used for the determination of the enzymatic activity and for the composition of the biosensors.

### 2.4. Enzymatic Activity Determination

#### 2.4.1. Total Polyphenoloxidase Determination (TPO)

To determine the enzymatic activity of total TPO, the reaction solution contained 3000 µL, with 750 µL of 100 mM sodium acetate buffer pH 5.0; 1800 µL and enzymatic extract of *M. colocasiae*; 150 µL of 2 mM H_2_O_2_ (Synth^®^); 300 µL of 0.03% ABTS (Sigma Aldrich^®^). For some extracts, it was necessary to dilute the sample in buffer.

The enzymatic activity was carried out in a spectrophotometer at 420 nm during 10 min [21,22].

#### 2.4.2. Laccase Activity Determination

For the determination of the enzymatic activity of the laccase, the reaction solution contained 3000 µL, with 850 µL of 100 mM sodium acetate buffer pH 5.0, 1800 µL and enzymatic extract of *M. colocasiae*, and 50 µL of commercial bovine catalase (Sigma Aldrich^®^). After 5 min of reaction with the catalase, 300 µL of 0.03% ABTS (Sigma Aldrich^®^) was added. For some extracts, it was necessary to dilute the sample in buffer. The enzymatic activity was monitored for 10 min in a spectrophotometer at 420 nm [21,22].

#### 2.4.3. Total Peroxidase Activity Determination

To determine total peroxidases, the following formula was used, subtracting both TPO and laccase activities, previously calculated [23].
Total peroxidases = Total TPO − Laccases,(1)

### 2.5. Development of M. colocasiae-Based Enzymatic Biosensor

The biosensor was composed of 70 mg graphite powder (Sigma Aldrich^®^) and 25 µL, 50 µL, 100 µL, or 200 µL of enzymatic extract containing TPO, called CP-25, CP-50, CP-100, and CP-200. The mixture was allowed to dry at room temperature for approximately 2 h (28 ± 2 °C). Subsequently, 30 mg of mineral oil was added to obtain a homogeneous paste. A quantity of the agglutinated paste was used to fill the cavity with 2 mm in diameter and 0.5 mm deep in the electrode holder, originating the enzymatic biosensors based on *M. colocasiae.*

Considering that three different production forms of enzyme extracts were tested, the biosensors were named based on the culture media, solid medium (MEA), liquid medium (ME), and liquid medium with addition of salts (MES), as shown in Table 1.

### 2.6. Electroanalytical Determination

Voltammetric analyses were performed on a PGSTAT^®^ model 204 potentiostat/galvanostat integrated with the NOVA2 software. 1^®^ (Metrohm Autolab, Utrecht, the Netherlands). The experiments were carried out in an electrochemical cell with a volumetric capacity of 10.0 mL, containing 2 mL of buffer solution and analyte, with a system of three electrodes consisting of biosensors, a Pt wire, and Ag/AgCl/KClsat, acting as working electrode, auxiliary electrode, and reference electrode, respectively. The carbon paste was manually renewed for each experiment, in order to ensure the result effectiveness and reproducibility.

For the working electrode conditioning, the cyclic voltammetry technique was used. The operating conditions were 10 successive cycles in 100 mM sodium acetate buffer pH 5.0 at 50 mV/s and a potential range from 0 to 1 V. Differential pulse voltammetry technique was chosen to perform the analyses of phenolic compounds. The operating conditions were pulse amplitude of 50 mV, pulse width of 0.5 s, and scan rate of 10 mV/s.

Electrochemical impedance spectroscopy (EIS) was performed to determine possible electrochemical changes in the proposed biosensor. The experiments were conducted in an electrochemical cell containing 2 mL of 5 mM K_4_ (Fe(CN)_6_). 3H_2_O in 0.1 MKCl. The EIS spectra were recorded at 0.2 V, in a frequency range from 10 kHz to 0.1 Hz with amplitude of 10 mV and 10 points measured per decade of frequency.

### 2.7. Biosensor Voltametric Analysis

In order to determine the best constitution for the biosensor, all mentioned variations in Table 1 were analyzed with the catechol substrate at 10 µM. The incubation time of the biosensor with the substrate was also tested before electrochemical determination.

In addition to the catechol substrate, catechin, rutin, 3-hydroxytyramine hydrochloride, hydroquinone, caffeic acid, and 10 µM gallic acid were analyzed by the optimized biosensor. The study of the optimum pH was performed with catechin 10 µM at CP-50ME in sodium acetate buffers pH 3.0 and 5.0, and with sodium phosphate pH 7.0 and 9.0, both at 100 mM.

To determine the possibility of reusing the same electrode in successive analyses due to the immobilization of enzymes on the electrode, a reuse test with 15 cycles using 30 µM catechin substrate in sodium acetate buffer pH 5.0 was performed. Precision and accuracy were performed in accordance with RDC N°. 166 [24], to be evaluated in the concentration in percentage of 80%, 100%, and 120% obtained at the end of 9 cycles.

Finally, a test of storage time of carbon paste at 4 °C in a refrigerator was carried out for a period of 28 days. As a substrate, catechin 10 µM was used in 100 mM sodium acetate buffer pH 5.0.

The data obtained were analyzed and treated by the Origin 8^®^ graphics software. Statistical analyzes were performed using the Minitab 19^®^ software. All experiments were carried out at room temperature (28 ± 2 °C) and in triplicate.

### 2.8. Biosensor Determination in Green Tea Samples

In order to quantify catechin and gallic acid in green tea samples by CP-50ME, two calibration curves were constructed for each pattern in the range from 1 to 60 µM in sodium acetate buffer pH 5.0, in addition to the limit of quantification and the limit of detection. It was also possible to determine precision and accuracy based on the calibration curves, with concentrations of 24.4 µM, 30.5 µM, and 36.6 µM.

The proposed biosensor was used in the determination of six different samples of green tea (*Camellia sinensis*) samples acquired in local market in Goiânia-GO, Brazil. One gram of extract was weighed and added in 10 mL of distilled water heated to 60–80 °C. Heating was interrupted by an ice bath after 2 min.

The green tea was prepared with 1 L of water at 40 °C to 80 °C, 10 g of green tea, and 70 g of sugar. It was submitted to an infusion for 15 min, then cooled to room temperature. Thereafter, the infusion and the yeast of commercial Acetobacter spp were added to erlenmeyers in triplicate at 25 °C for 15 days; this time was necessary for the yeast to ferment the whole sugar content.

### 2.9. Chromatographic Conditions

HPLC system (Model-LC-20AT, Shimadzu Corporation, Kyoto, Japan) equipped with low pressure quaternary gradient pump, column oven, autosampler, and PDA detector was used for analysis. Chromatography data was processed by LabSolution software (Shimadzu, Kyoto, Japan). Analysis was performed on Reverse phase C18 ACE^®^ (250 × 4.6 mm i.d., particle size 5 μm) column coupled with security guard column (4 × 3 mm, Phenomenex^®^) at room temperature. The mobile phase was composed of 0.1 % phosphoric acid (solvent A) and methanol (solvent B) in gradient elution mode as follows: 0–1 min 85% of solvent A; 1–4 min linear gradient to 70% of solvent A; 4–5 min 70% of solvent A; 5–6 min linear gradient to 85% of solvent A; 6–20 min 85% of solvent A to re-equilibrate the system. The mobile phase was degassed before pumping into the HPLC system at a flow rate of 1.0 mL/min. Injection volume of 50 μL was used. The wavelength monitored was 280 nm.

### 2.10. Total Phenols Spectrophotometric Determination

The Folin-Ciocalteu (FC) spectrophotometric method was used to determine total phenolic compounds in green tea samples [12]. Each aliquot of 50 µL of green tea sample at a concentration of 1% was placed in a test tube containing 1 mL of distilled water and 250 µL of the FC reagent. After 5 min, 750 µL of a 20% Na_2_CO_3_ solution and 2950 µL of distilled water were added. The mixture was incubated in the absence of light for 60 min; afterwards, the absorbance was measured in a spectrophotometer at 765 nm, using the blank solution as a reference. The quantification of phenolic compounds in green tea samples was carried out in triplicate and expressed by means of gallic acid equivalents in µM, from a calibration curve obtained under the same conditions for sample analysis [24].

## 3. Results and Discussion

### 3.1. Biosensor Production

Different culture media were tested for *M. colocasiae*, in order to obtain crude enzymatic extracts containing high TPO activity. Biosensors with different concentrations of crude enzymatic extract were tested with the catechol substrate (Figure 1A). From these results, the best analytical responses for electrochemical reduction of the substrate with the biosensors of each culture medium, the CP-100MEA, CP-100MES, and CP-50ME were selected. Subsequently, the optimization of the incubation time was performed with the aforementioned three types of biosensors to analyze the enzymatic reaction to oxidize the substrate, followed by the electrochemical reduction (Figure 1B).

The enzymatic conditions and the interaction with the CP can generate variations in relation to the electrochemical responses; these were optimized, presenting better results in the CP-100MEA. The enzymes interact specifically with the phenolic substrate by redox mechanisms (Appendix A). The enzymatic activity generates electrochemical responses based on the amount of extract added and the time of enzymatic interaction, the amount interferes with the resistivity of the electrode due to the protein properties of the enzyme that are not very electroactive and can reduce or block the signal.

The MES medium was expected to present the best responses, since it represented a complex and rich medium for the production of TPOs; however, it was not observed for this strain of *M. colocasiae*. Some factors can modify the enzymatic behavior, such as water excess, presence of halides, organic salts, inorganic salts, and metal ions such as cobalt, manganese, and copper present in the active sites of some enzymes. Moreover, interfering or competing with the active site, in order to inhibit or decrease the enzymatic action, can justify the low enzymatic production of the MES medium (Figure 1A) [25].

The salts added in the MES copper sulphate, magnesium sulphate, and manganese sulphate, when solubilized, could form a sulphate anion by denaturing the protein, or by changing the pH by inactivating the enzyme, similarly with dipotassium phosphate that can dissociate and form the hydrogen phosphate anion, modifying the morphological and catalytic enzymatic conformation.

As shown in Figure 1, the best biosensor was the CP-50ME with an enzymatic reaction time of 2 min and 30 s, it showed 3% deviation from analytical response value in relation to CP-100ME, the CP-25ME obtained 65%, and the CP-200ME presented 48% of analytical response for the substrate in relation to the best sensor. CP50-ME biosensor was standardized under ideal conditions for the other tests, due to its better catalytic response. The CP-25 in MES and MEA did not show significant results in relation to unmodified CP.

The CP-50ME biosensor was studied in relation to changes in the carbon paste surface by electrochemical impedance spectroscopy (EIS), for this purpose, an unmodified sensor was also tested (Figure 2). From the Randles electrical circuit, the solution resistance (Rs), the charge transfer resistance (Rct), and the system pseudo-capacitance (C) were obtained, with values of Rs = 63.7 KΩ, Rct = 23.7 KΩ, and C = 1.02 µF for the carbon paste without modification and Rs = 58.6 KΩ, Rct = 28.6 KΩ, and C = 2.09 µF for the biosensor.

EIS showed that charge transference resistance and pseudo-capacitance in the biosensor were greater in the biosensor in comparison to the unmodified electrode. That implies that the enzymes were likely present in the electrode surface and are promoting greater charge accumulation in this site.

### 3.2. Biosensor Performance

Standard substances were used to test the analytical capacity of the proposed biosensor against different phenolic compounds, such as catechin, epicatechin, catechol, 3-hydroxytyramine hydrochloride, caffeic acid, gallic acid, and hydroquinone (Figure 3).

Intermolecular and intramolecular factors can modify the reactivity of the tested patterns such as the number of hydroxyl moieties in the phenolic compounds and the water solubility of these species. Phenolic compounds form hydrogen bonds when they come in contact with water forming H_3_O^+^; the larger the molecule and the greater the amount of carbon, the more insoluble in water it becomes. In addition, ortho positions accompanied with nitrogen may show solubility below para and meta ones [26].

As observed in Figure 3, catechin and epicatechin presented similar responses to the proposed biosensor, as observed in the literature [9,27]. Catechins show peak reversibility as shown in the literature using CV, as well as the catechol group phenolic compounds, which were orthokinones. This behavior can be justified by these compounds being stereoisomers, which differ by the spatial conformation of cresol, with 5 hydroxyls and water solubility of 0.018 g/mL [28,29].

Regarding catechol and rutin, both species showed similar results. Catechol has 2 hydroxyl groups and a solubility of 0.43 g/mL, whereas, rutin has 9 hydroxyl groups and solubilizes at 0.1 g/mL [30,31].

Other compounds that presented similar data were 3-hydroxytyramine hydrochloride, gallic acid and caffeic acid; 3-hydroxytyramine hydrochloride has 2 hydroxyl groups and a solubility of 0.6 g/mL [24]; gallic acid has 4 hydroxyls and a solubility of 0.015 g/mL [25], caffeic acid has 2 hydroxyls and a solubility of 0.25 g/mL [26].

Finally, hydroquinone, which presented the worst response with the biosensor, is a quinol and has two hydroxyls in the para position and a solubility of 0.059 g/mL. Even though it is an isomer of catechol, the position of the hydroxyl in the ring completely changes the affinity of the enzyme for the compound completely [32,33]. From these results, catechin was used as a standard substance to determine the optimal performance parameters of the CP-50ME biosensor.

The evaluation of optimal pH was performed (Figure 4) and pH 5.0 exhibited the best result which is in agreement with previous publications for fungal enzymes [19,34]. The electrochemical measurements were performed at this pH, maintaining the ideal conditions for optimizing the enzymatic activities and the active site substrate interaction.

Still in relation to the performance of the biosensor, successive measurements with the same electrode were performed at a concentration of 30 µM (Figure 5). The reuse showed a catalytic decay of 39% at the end of 15 cycles in the determination of catechin. Evaluating the percentage of activity and decay of the biosensor together with the variation coefficient, the 5th cycle presented 83 ± 11.16%, in the 10th cycle 70 ± 12.13%, and in the 15th cycle 61 ± 13%, which on average generates a variation of approximately 13% every 5 cycles. The satisfactory reuse will be until the 8th cycle, which presents a 20% decay, corroborating data from the literature for biological samples [35].

Based on the equation cited to perform the electrocatalytic activity of the laccase in biosensors, it was possible to measure the electrocatalytic activity of the enzyme where the maximum current obtained in the catechin concentration was divided by the Michaelis-Menten constant obtained by the activity enzymatic analysis performed on a spectrophotometer, with the result obtained for laccase 1100, 830 U L^−1^, and the I correspond to −6.73 µA [36]. The equation is based on:I = Imax[S]/[S] + Kapp(2)

The response obtained was 17.86 µM^−1^ of electrocatalytic affinity, corroborating to Figure 2, which emphasizes that catechin has a greater efficiency for redox activity at a concentration of 30 µM.

The carbon paste modified with the crude enzymatic extract was also stored for the construction of the biosensor (Figure 6). Storage was performed at 4 °C and the decay of the enzymatic activity of the biosensor was monitored for 28 days.

The first day of storage showed less response compared to the 7th, 14th, and 21st days; this fact can be explained by the physical adsorption being still incomplete on the first day, demonstrating that the graphite has an enzymatic incorporation time [37].

### 3.3. Green Tea Sample Analysis

With the adjustments of enzyme extract concentration in the manufacture of the biosensors, time of enzymatic reaction between biosensor and analyte, optimal pH, reuse cycles, and storage time allowed the calibration curves to be carried out for catechins and gallic acid (Figure 7). According to data in the literature, commercial samples of green tea (*Camellia sinensis*) have a large amount of catechin and gallic acid [38,39].

Based on resolution of the collegiate board (RDC) 166 [24] which provides for analytical methods, it infers that precision evaluates the proximity between the results obtained by calculating the relative standard deviation (DPR), while the accuracy is achieved through the degree of acquiescence between the individual results of the method expressed by the percentage recovery as shown in Table 2. Cycles done in triplicate in 80%, 100%, and 120% from the tests performed in triplicates describe that the proposed method was precise and exact, in line with the concentration in µM.

The HPLC method was used as a standard for the detection of catechin and gallic acid (Appendix A), making it possible to identify the presence of gallic acid in samples A, B, and C, coeluting with other compounds in the samples. Catechin was not identified in any sample. Given the sensitivity of the biosensor to catechin and gallic acid, it is inferred that the proposed biosensor was able to quantify and identify the patterns by the current more effectively than the method used in HPLC.

Works developed by the research group corroborated the results obtained with the biosensor developed in this research with other immobilized polyphenol oxidases electrodes for the detection of catechol group phenolic compounds, catechins, and galactic acid [5]. The results presented in Table 3 present compiled biosensors in the literature with the one proposed, comparing the detection limit and electrochemical techniques used.

Table 4 presents the voltammetric method (developed biosensor) and the standard spectrophotometric method for detecting total phenolic compounds, called Folin-Ciocalteu assay. In samples A and C, the gallic acid is agreement with the Folin-Ciocalteu method. In sample B the amount of gallic acid is less than the spectrophotometric method since it quantifies the total phenols and not only gallic acid. In samples D and F, the amount of total phenols is higher in relation to the catechin concentration. Sample E showed a higher concentration of catechin in relation to the Folin-Ciocalteu method.

In view of the better responses of samples D, E, F presented by the tests of total and electrochemical phenols, kombuchas called samples G, H, and I were fermented for the best results, as a comparative method for normal infusion and fermented drink [11].

As statistics inferred from coherent data, we evaluated the variation between the spectrophotometric data, Folin-Ciocalteu assay, with the proposed electrochemical method. The Bonett statistical method has a 95% confidence interval and the significance level α = 0.05. Gallic acid and catechin showed *p* > 0.05, rejecting a hypothesis (H0), since this value measures the strength against H0 [44].

An ANOVA analysis of variance was performed with the concomitance with the Bonnet tests, being an accurate test in the distributions and continuations, being considered a reliable test, in comparison with the spectrophotometric Folin-Cicalteu methods for detection and quantification of total phenols.

Studies have shown that the enzymatic biosensor of *M. colocasiae* presented relevant results for a new analytical tool with low cost, high sensitivity, and reproducibility, guiding the group’s focus on research with real samples under usual conditions for the detection of phenolic compounds, optimizing the control of quality in green tea, aiming at the development of new electrochemical detection devices [5].

## 4. Conclusions

The enzymatic crude extract from the phytopathogenic basidiomycete *M. colocasiae* was used on the development of biosensor for total polyphenol determinations. The optimized biosensor was applied in the detection of phytochemical markers gallic acid and catechin for the quality control of green tea and kombucha.

The analytical figures of merit, including precision, accuracy, sensitivity, linearity, and reuse signal stability reinforce the expected performance for fungal polyphenoloxidases to produce enzymatic biosensors for total phenol content determinations.

The easy production, low cost, and quick analysis are attractive traits of the proposed biosensor as an alternative tool for the quality control of foodstuff and beverage analysis.

The results observed for teas and kombucha did not evidence any positive, neither negative effect of fermentation on total phenol content.

## Figures and Tables

**Figure 1 biosensors-11-00091-f001:**
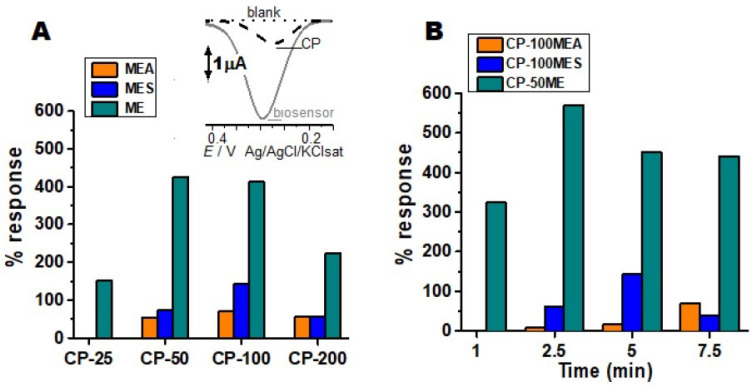
Optimization of biosensors with crude enzymatic extracts. Control: Non modified carbon paste (**A**) enzymatic extracts obtained from solid medium (MEA), minimal liquid medium (ME), and complex liquid medium (MES) in different proportions (25, 50, 100 or 200 µL in 70 mg of graphite) in the construction of biosensors, tested with the catechol substrate at 10 µM. Inset: CP-50ME and bare CP in the analysis of the catechol 10 µM. (**B**) Different enzyme reaction times of the biosensors in the 10 µM catechol solution, before electrochemical analysis. All tests were performed in 100 mM sodium acetate pH 5.0 buffer.

**Figure 2 biosensors-11-00091-f002:**
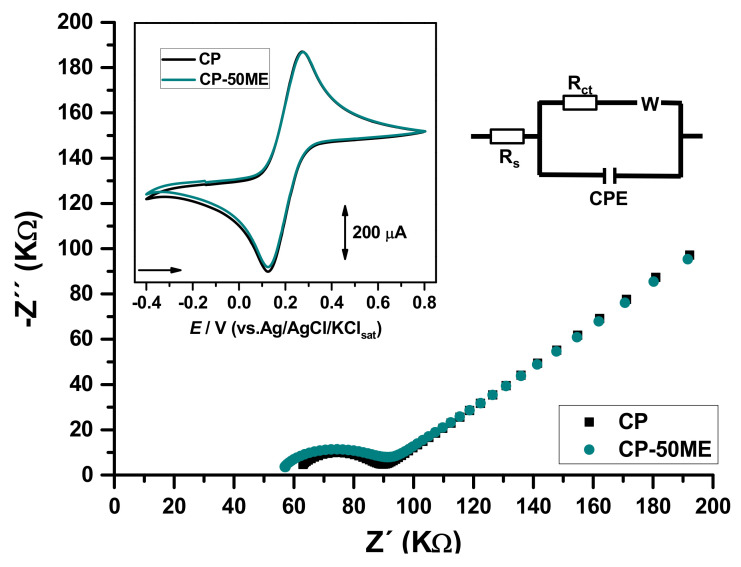
Nyquist plots for CP and CP-50ME. Inset: Cyclic voltammograms at. All analyses were carried out in 0.1 M KCl solution containing 50 mM K_4_(Fe(CN)_6_)/K_3_(Fe(CN)_6_).

**Figure 3 biosensors-11-00091-f003:**
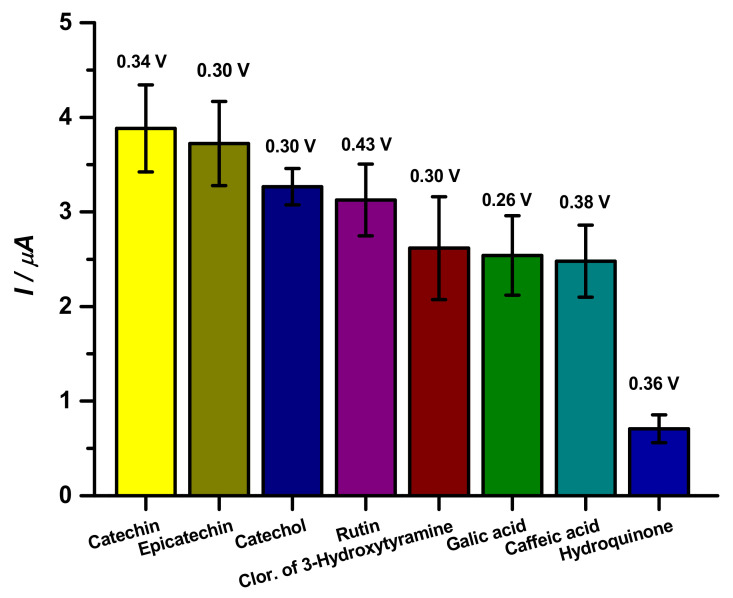
Analysis of phenolic compounds at 10 µM by CP-50ME in sodium acetate buffer pH 5.0 after 2 min and 30 s of enzymatic reaction.

**Figure 4 biosensors-11-00091-f004:**
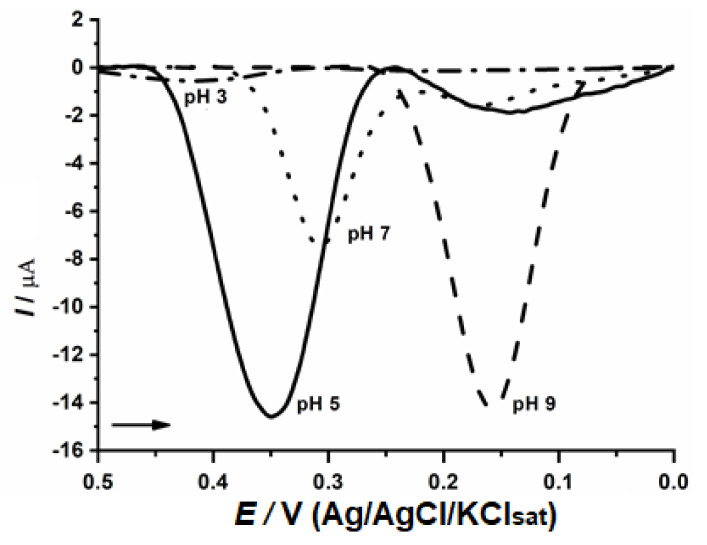
Determination of optimal pH with CP-50ME in 40 µM catechin solution in sodium acetate buffer pH 3.0 and 5.0 and sodium phosphate buffer pH 7.0 and 9.0, after 2 min of enzymatic reaction.

**Figure 5 biosensors-11-00091-f005:**
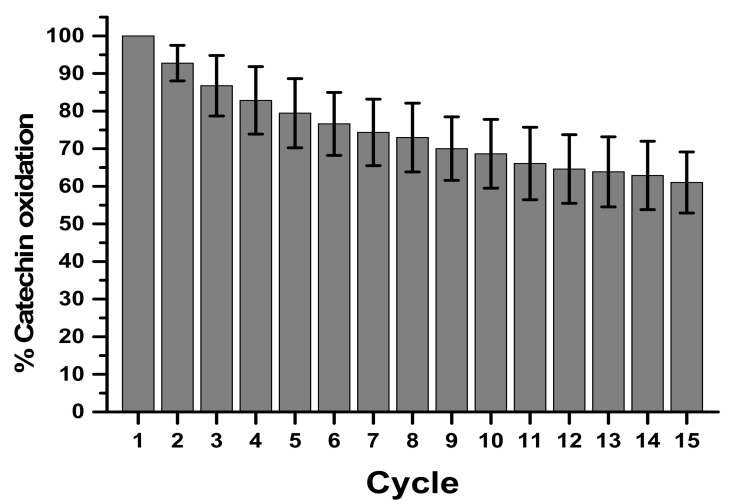
Reuse of CP-50ME after 15 successive cycles in the determination of catechin at 30 µM in sodium acetate buffer pH 5.0 after 2 min of enzymatic reaction. The average provided corresponds to three determinations.

**Figure 6 biosensors-11-00091-f006:**
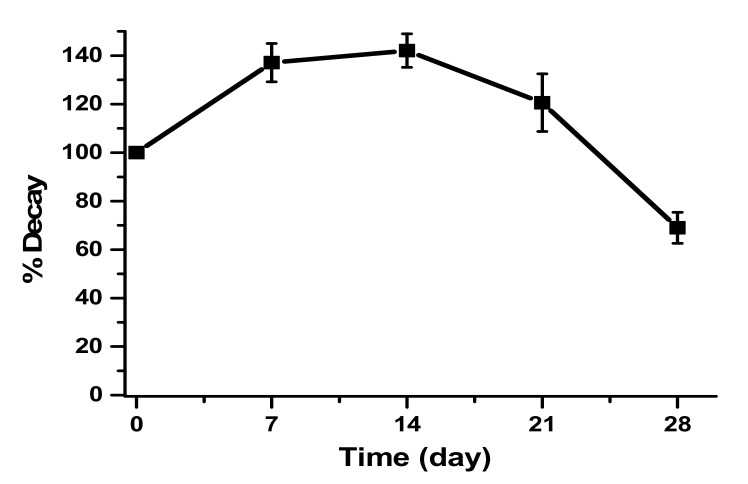
Storage for 28 days at 4 °C of CP-50ME in the determination of catechin at 30 µM in sodium acetate buffer pH 5.0 after 2 min and 30 s of enzymatic reaction.

**Figure 7 biosensors-11-00091-f007:**
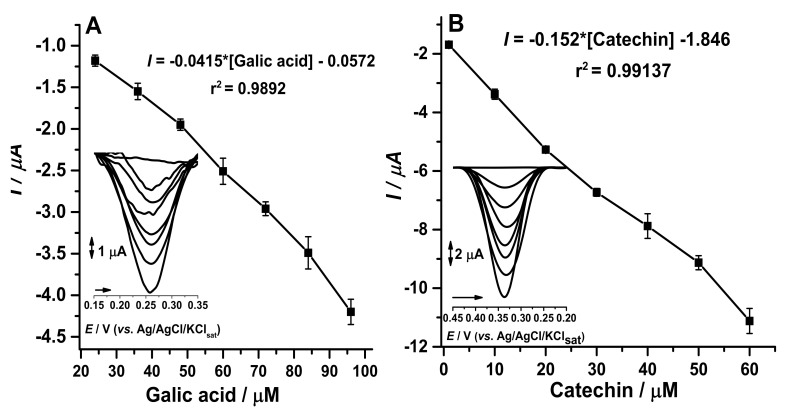
(**A**) Calibration curve with gallic acid from 1 µM to 60 µM at CP-50ME with LD = 0.14 µM and LQ = 0.42 µM. (**B**) Calibration curve with catechin from 1 to 60 µM at CP-50ME with LD = 0.12 µM and LQ = 0.38 µM. Both performed in 100 mM sodium acetate buffer pH 5.0 after 2 min and 30 s of enzymatic reaction.

**Table 1 biosensors-11-00091-t001:** Biosensor labeling based on culture medium and enzymatic extract content.

	Enzymatic Extract Content			
Culture Medium	25 µL	50 µL	100 µL	200 µL
MEA *	CP-25MEA	CP-50MEA	CP-100MEA	CP-200MEA
ME **	CP-25ME	CP-50ME	CP-100ME	CP-200ME
MES ***	CP-25MES	CP-50MES	CP-100MES	CP-200MES

* MEA = malt and agar extract; ** ME = malt extract; *** MES = malt extract and salt addition.

**Table 2 biosensors-11-00091-t002:** Precision and accuracy of the CP-50ME biosensor in 100 mM sodium acetate buffer pH 5.0 after 2 min and 30 s of enzymatic reaction in the determination of catechin. The average provided corresponds to three determinations.

Concentration (%)	Concentration (µM)	Average (µM) ± Standard Deviation	Relative Standard Deviation (%)
80	24.4	23.12 ± 1.94	8.38%
100	30.5	30.31 ± 2.32	7.64%
120	36.6	35.64 ± 2.73	7.66%

**Table 3 biosensors-11-00091-t003:** Comparison of the results obtained with the biosensor developed in this research with other immobilized polyphenol oxidases electrodes for the detection of catechin, gallic acid, and phenolic compounds in green tea samples.

Enzyme Source	Phenolic Targets	Method	Linear Range (µmol L^−1^)	Limit of Detection (µmol L^−1^)	Reference
*M.colocasiae*	Catechin	DPV	1–60	0.12	This work
	Gallic acid		3–60	0.14	
*M.colocasiae*	Catechol	DPV	50–300	0.17	[5]
Agaricus bisporus	Polyphenols from green tea	CV and amperometric	0–2.4	0.3	[40]
Agaricus bisporus	polyphenols	Amperometric	0.5–101	0.15	[41]
Biomimetic	Catechin	CV and amperometric	2–20	1.5	[42]
	Gallic acid		50–200	15.3	
Gallic acid	Catechin	DPV	0.1–2.69	0.017	[43]

**Table 4 biosensors-11-00091-t004:** Analysis of green tea samples with CP-50ME in 100 mM sodium acetate buffer pH 5.0 after 2 min and 30 s of enzymatic reaction.

Samples	*E*_pc_ (V) (*n* = 3)	*I*_pc_ (µA) ± Standard Deviation (*n* = 3)	Catechin Equivalent (µM) ± Standard Deviation (*n* = 3)	Gallic acid Equivalent (µM) ± Standard Deviation (*n* = 3)	Folin-Ciocalteu Gallic Acid Equivalent (µM) ± Standard Deviation (*n* = 3)
A	0.26	−1.6 ± 0.23	-	37.09 ± 3.8	36.74 ± 4.19
B	0.25	−0.62 ± 0.12	-	13.55 ± 2.87	36.24 ± 6.40
C	0.26	−1.24 ± 0.10	-	28.66 ± 2.47	23.24 ± 4.61
D	0.34	−2.54 ± 0.23	4.44 ± 1.5	-	14.89 ± 2.35
E	0.34	−7.82 ± 0.40	39.2 ± 2.62	-	22.47 ± 1.02
F	0.33	−2.90 ± 0.54	6.91 ± 3.59	-	23.27 ± 2.70
G	0.33	−2.54 ± 0.23	1.59 ± 0.24	-	41.43 ± 3.4
H	0.33	−7.81 ± 0.39	2.03 ± 0.32	-	30.69 ± 4.17
I	0.32	−2.89 ± 0.54	3.65 ± 0.47	-	21.54 ± 1.06

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
