# Peer review of "Enzymatic Electroanalytical Biosensor Based on *Maramiellus colocasiae* Fungus for Detection of Phytomarkers in Infusions and Green Tea Kombucha"

_biosensors, 2021, doi:10.3390/bios11030091_

Round 1

Reviewer 1 Report

Batista et al described a enzymatic biosensor we  the determination of 
catechin and gallic acid. After optimizing the sensor, they applied it real samples, namely green tea and kombucha samples. Moroever, they compared the developed biosensor with HPLC and spectroscopofic methods. However, it can be published after minor revision.

1)  y -axis of the DPVs and CV including insets should be demonstrated.

2) A table, comparing the developed biosensors and already published sensors  (Linear range, limit of detection, etc for catechin and gallic acid),  should be prepared.

3) Calibration curves ( Fig. 7 for gallic acid and catechin) should be demonstrated with error bars.

Author Response

Response to Reviewers

We are very grateful for the collaboration of the biosensors mdpi reviewers. The work has improved a lot after your contributions. Below are the corrections as requested:

Reviewer 1

  • The X and Y axes were included as requested by the reviewers.
  • Table was prepared and inserted in the results called table 3.
  • The error bar was inserted in figure 7 as requested by the reviewer.

Reviewer 2 Report

Dear Author,

Journal: Biosensors (ISSN 2079-6374)

Manuscript ID: biosensors-1110671

Type: Article

Number of Pages: 16

Title: Enzymatic electroanalytical biosensor based on Maramiellus colocasiae fungus for detection of phytomarkers in infusions and green tea kombucha

Authors: Erica Batista , Giovanna Mello , Livia Sgobbi , Fabio Machado , Isaac Yves Macedo , Emily Moreno , Jeronimo Oliveira Neto , Paulo Sérgio Scalize , Eric Gil *

General comments:

I did not notice a difference between these two versions of the paper.

However, some details of the work need to be improved.

Please find below my comments which need to be addressed.

  1. The introduction is very poorly written. There are many similar papers on the topic of biosensors for antioxidant in green tea or use of Maramiellus colocasiae fungus as bioactive sensor material, but author did not mention any of them (not even your own), and that is the topic of the paper. Recommendation: Write the introduction again with a brief overview of previous achievements and trends biosensors for green tea or use of Maramiellus colocasiae fungus. The number of references speaks in favours of a poorly Introduction.

I think the motivations for this study need to be made clearer.

The objective is clearly defined in the last sentence of the second paragraph. (line 60-66)

 Methods: The experimental apparatus is quite standard, and is appropriate for the study. I don’t think any additional experiments are necessary to validate the results presented here.

Results and discussion:

EIS data are shortly described in the text.

There is not a single cyclic voltammogram for phenolic compounds.

Literature: Check all reference. REF 10 and 11…check in the text. Sort references by the appearance in the text.

After corrections modify conclusion if you feel you need.

Please find attached a version of our manuscript.

The recommendation for the editor:

Accept after major revisions.

Sincerely

Author Response

Response to Reviewers

We are very grateful for the collaboration of the biosensors mdpi reviewers. The work has improved a lot after your contributions. Below are the corrections as requested:

Reviewer 2

  • The introduction and reference of the research group that addresses a study on M. colocasiae has been rewritten.
  • The aim of the EIS study was only to briefly address the impedance diference between the modified and unmodified electrode, which was the reason for the very brief discussion of this method.
  • References 10 and 11 have been modified with a change to the introduction.

Round 2

Reviewer 2 Report

Dear Editor,

Thank you for inviting me to re-review the manuscript for:

Journal: Biosensors (ISSN 2079-6374)

Manuscript ID: biosensors-1110671

Type: Article

Number of Pages: 16

Title: Enzymatic electroanalytical biosensor based on Maramiellus colocasiae fungus for detection of phytomarkers in infusions and green tea kombucha

Authors: Erica Batista , Giovanna Mello , Livia Sgobbi , Fabio Machado , Isaac Yves Macedo , Emily Moreno , Jeronimo Oliveira Neto , Paulo Sérgio Scalize , Eric Gil *

The revised manuscript - the authors significantly improved the previous version of the paper.

The recommendation for the editor:

This revised manuscript can be accepted in its current form.

Sincerely

Author Response

Reviewer comments: The revised manuscript - the authors significantly improved the previous version of the paper.

The recommendation for the editor: This revised manuscript can be accepted in its current form.

We appreciate the effort of the reviewer to revise carefully, our paper.